# I like you better when you are coherent. Narrating autobiographical memories in a coherent manner has a positive impact on listeners' social evaluations

**Lauranne Vanaken**[1]*, **Patricia Bijttebier**[2], **Dirk Hermans**[1]

**1** Centre for the Psychology of Learning and Experimental Psychopathology, KU Leuven, Leuven, Belgium,
**2** School Psychology and Development in Context, KU Leuven, Leuven, Belgium

* lauranne.vanaken@kuleuven.be

## Abstract

**Data Availability Statement:** The pre-registration of the study can be found on AsPredicted (https://aspredicted.org/y3d8n.pdf). All data files are available from the Open Science Framework

### Introduction

We all have stories to tell. The stories that prevail in our conversations frequently concern significant past personal experiences and are accordingly based on autobiographical memory retrieval and sharing. This is in line with the social function of autobiographical memory, which embodies the idea that we share memories with others to develop and maintain social relationships. However, the successful fulfilment of this social function is dependent on phenomenological properties of the memory, which are highly inter-individually different. One important individual difference is memory coherence, operationalized as narrative coherence. The objective of this study was to investigate the impact of memory coherence on the social evaluations of listeners. We hypothesized that being incoherent in the sharing of autobiographical memories, would evoke more negative social evaluations from listeners, in comparison to coherently sharing autobiographical memories.

### Methods

In a within-subject experimental study, 96 participants listened to four pre-recorded audio clips in which the speaker narrated about an autobiographical experience, in either a coherent or an incoherent manner.

### Results

Results were in line with our hypotheses. Participants showed more willingness to interact, more instrumental support, more positive feelings, more empathy and more trust towards those narrators who talked in a coherent manner about their autobiographical memories, as compared to those that talked in an incoherent manner. Negative feelings in the listener were evoked when the speaker talked incoherently, but especially when it concerned a positive memory.

database (DOI 10.17605/OSF.IO/534NQ). On this OSF page, a duplicate of the AsPredicted pre-registration is also included.

**Funding:** This article was funded by Fonds Wetenschappelijk Onderzoek (FWO) Research Project G070217N (PI: DH) and by a KU Leuven Research Council Grant PF/10/005 (DH). The funders had no role in study design, data collection and analysis, decision to publish, or preparation of the manuscript.

**Competing interests:** The authors have declared that no competing interests exist.

## Discussion

Results can be explained in terms of a reduction in the attraction effect when effortful processing is increased, which is in line with the dual processing theory of impression formation. Another explanation involves the idea that coherence is necessary to establish truthfulness in communication. The clinical relevance of these findings is further illustrated in light of the relation between social support and psychological well-being.

## Introduction

We all have stories to tell. The stories that are frequent topic of conversation concern significant personal experiences from our past, and are accordingly based on autobiographical memory retrieval [1]. It is the autobiographical memory that serves to recollect past personally experienced events and to integrate those into meaningful narratives [2]. Hence, it is evident that one of the three main functions of autobiographical memory is a social function, which embodies the idea that we share memories with others to develop and maintain social relationships [3–6]. In addition to this self-in-relation function (creating and nurturing social bonds by remembering and sharing past experiences with others), there is evidence for a self-definition function (using past experiences to form a sense of self and identity) and a self-regulation function (directing future behaviour based on past experiences) [7–9].

The social sharing of autobiographical memories is shown to take place very frequently [10]. Research suggests that we share up to 90% of memories of emotional events on the same day and even within a couple of hours after experiencing them [1]. Even more crucial, besides frequent use, the sharing of memories serves an important function [11] . It namely facilitates the development and maintenance of social relationships over time and thereby fulfils our primary need to belong [12]. For example, Alea and Bluck [11] found that indicators of intimacy, such as warmth and closeness to others, increased after narrating about personally experienced relationship events (e.g. their own vacation) as opposed to talking about non-autobiographical vignettes (e.g. another couple's vacation). The fulfilment of our need to belong, or our ability for human connection, has major implications for our psychological well-being, as is extensively evidenced by literature indicating that a stable social network is essential for ensuring physical and mental health [13,14]. It has been repeatedly shown that good social support enhances resilience to stress and is a protective factor for psychopathology [11]. Likewise, a lack of social support has been associated with a decrease in psychological well-being and a higher likelihood of developing feelings of loneliness, symptoms of depression [15, 16,17]. In sum, sharing personal memories allows us to develop a social network, thereby providing us with a sense of belonging, which is vital for our psychological well-being.

In this process of storing, retrieving and sharing autobiographical memories, individual differences (e.g. accuracy, specificity, emotional tone) have been observed, that are able to impact the extent to which the social function is served and consequently our well-being [18–20]. One example can be found in the interactional model of depression, in which Coyne highlighted the major role that dysfunctional social behaviour can play in maintaining psychopathology [21]. In his research, depressed persons expressed their negative thoughts, feelings and memories in such a hopeless and self-blaming way that feelings of guilt, hostility and annoyance were aroused in others around them. This led the patients and their surroundings to get embedded in an increasingly negative spiral of social interactions, causing social rejection over time, which is a major factor in maintaining depressive symptoms [22]. This study is seen as

important and relevant here since it was one of the first to show how characteristics of communication (narrative styles) can impact social behavior and even the social maintenance of symptoms of psychopathology.

Another example of an individual difference variable in the process of retrieving and sharing memories that is gaining attention in the field is *autobiographical memory coherence*, operationalized as *narrative coherence* [23]. Coherence of autobiographical memories is mostly assessed using narratives (i.e. a written or spoken account of a personal experience/autobiographical memory) [24]. A coherent personal memory is defined as one that makes sense to a naive listener, not just in terms of understanding when, where, and what event took place, but also with respect to understanding the meaning of that event [24].

Coherence has been described as a multidimensional concept, that entails a contextual dimension (are the where and when of the event specified?), a chronological dimension (is the story told in chronological order?), and a thematic dimension (does the narrator have insight in the experience, comes to a resolution, reaches closure, links the event with other events in the past or possible future events?) [24]. The concept has been predominantly investigated in the face of negative life events (trauma) and with regard to the risk of developing symptoms of depression [25–27] and PTSD, in multiple populations like terror attack survivors [28] and persons who recently got divorced [29]. In these studies, those who were able to construct a coherent story about their negative life experiences or trauma, appeared to have better mental health than those who were not able to do so. Summarized, coherence is a characteristic of memories that has been positively related to psychological well-being and negatively related to psychopathology [30,31].

However, it is not clear yet what the precise mechanisms are that underlie the relation between autobiographical memory coherence and psychological well-being. Concerning this question, it is suggested that the aforementioned social function of autobiographical memory might be one of the mechanisms at stake. Namely, we suggest that good social functioning mediates the association between coherence and mental health. Hence, in our study we will focus on the effect that memory coherence of a speaker has on listeners' social evaluations. It is hypothesized that when someone is not coherent in the sharing of personal memories, this will disturb the social function of autobiographical memory. We believe that memory coherence is part of an individual narrative style that remains relatively stable over time and over situations [32]. We hypothesize that narrating incoherently can cause feelings of annoyance or confusion in the listener, and that being incoherent over and over again could set in motion a negative spiral of social interactions, possibly causing social rejection with time. This means that social support could diminish when the speaker remains incoherent, which then may impact psychological well-being in a negative way [12–14], especially in the face of adversity or trauma (risk factor for psychopathology). Similarly, telling a coherent story could be reinforced by receiving positive social feedback and support, nurturing social bonds and satisfying the need to belong over time, which could then improve well-being (protective factor for psychopathology).

Preliminary support for the mediating role of good social functioning between memory coherence and mental health is found in a study of Waters and Fivush [31]. These authors showed that the ability to create a coherent narrative is related to having positive social relationships (defined as having a positive appraisal of one's social relationships and functioning, measured by perceived social support, social well-being and generativity). Moreover, Burnell, Coleman, and Hunt [33] compared the types of social support that were experienced by veterans with a coherent, reconciled or incoherent narrative. They showed that veterans with a coherent narrative perceived communication with family to be pleasant, and that they experienced societal opinion to be more positive. In contrast, veterans with incoherent narratives found communication unsatisfactory, feeling prevented to talk about their war memories,

because of perceiving both their social circle of family and friends as well members of society to be less interested and misunderstanding. Also, veterans with incoherent narratives did mention the need for communicating and managing their memories, however the social support to do so was not available or insufficient. A personal and societal support network that is open to communication seems vital to make meaning of (traumatic) memories [33].

Summarized, there is quite extensive research on the social function of autobiographical memory within the framework of the importance of social support for mental health. However, the impact of the frequent sharing of memories on social evaluations of the listener seems to be left untouched. This leads us to the main question of this study, which concerns the investigation of the impact of autobiographical memory coherence of the speaker on social evaluations of the listener.

In this experimental study, we manipulated the coherence of the story of the speaker, to examine its effects on a range of social evaluations from the listener (the participant). We predicted that participants would respond with more willingness to interact, a higher degree of emotional and instrumental support, more positive and less negative feelings, more empathy and trust, when listening to coherently narrated upon memories, in comparison to listening to incoherent memories. Since memory coherence is mostly assessed in the form of narratives about high-impact positive and negative memories [27,34], valence of the memories was assessed secondarily to explore possible interactions with coherence [35].

## Methods

### Participants

A total of 96 adults between the ages of 19 and 40 ($M$ = 21.06, $SD$ = 3.17) participated in the study, of which 84 (87.5%) were female and 12 (12.5%) were male. Our sample in this study was very homogeneous, consisting of mostly young white female students, with only a couple of outliers of people higher than 25 years in age. All of them were Belgian and indicated Dutch as their mother tongue or indicated actively speaking it. Participants signed up via the Experiment Management System (EMS) of the KU Leuven, so most of them were university students. All gave written informed consent before the start of the study and received either one course credit or remuneration (€8) for their participation. The study was approved by the KU Leuven Social and Societal Ethics Committee (G—2018 03 1175).

### Material and measures

**Narratives.** We created 16 narratives (in Dutch) based on themes that are very common in this sample and representative for self-reported events with high emotional impact. We wrote the stories, based on our extensive experience collecting and coding hundreds of narratives in similar samples of our own studies (in prep), and investigating event types in similar work [25,26,36] We used 4 positive (graduation, falling in love, birthday party, travelling) and 4 negative themes (suicide of a friend, divorce of parents, passing away of a grandparent, end of a relationship), about which we wrote a coherent and an incoherent story each. Subsequently, two colleagues specialized in the field independently and blind for condition coded these 16 narratives for coherence using the Narrative Coherence Coding Scheme (NCCS; [24]).

This coding system evaluates narratives on 3 separate dimensions (score 0–3) that are summed up to entail total memory coherence (score 0–9) (See S1 Appendix for scoring criteria, adopted from Reese et al., 2011, p. 436). Low scores on these dimensions indicate incoherent narratives, whereas higher scores show a more coherently constructed narrative. All dimensions need to be present to get a very high score. However, intermediate scores can be

reached in different ways (e.g. low context, high chronology, low theme or high context, low chronology, low theme), so there is a certain compensation or interchangeability in the measure. The first dimension is 'context', which refers to how the narrator orients the event in time and space. If the narrator does not provide any information about time or place, score 0 is assigned. If there is partial information, meaning that only the time in which or the location where the event took place are mentioned, at any level of specificity, a score of 1 is assigned (specific time e.g.: when I was 7 years old, nonspecific time e.g.: when I was a child, nonspecific place e.g.: when I was abroad, specific place e.g.: at my grandmother's house). A score of 2 is assigned when both time and location are provided, but no more than one dimension is specific. When time as well as location are mentioned both specifically, a score of 3 is given. The second dimension is 'chronology', which refers to whether the narrator describes the components of the events along a (chrono)logical timeline. If the narrator describes less than 3 actions of which the total event consisted (very short narratives like: when my mother passed away), a score of 0 is assigned. If the narrator describes more than 3 actions but fewer than half can be ordered on a timeline by a naïve listener, score 1 is given. When more than half of the actions can be ordered on a timeline by a naïve listener, score 2 is assigned. Score 3 is given when almost all actions can be ordered on a timeline and the narrator uses words (e.g.: right before, after an hour, the next day) to mark the temporal order of the actions. The third dimension is 'theme', which refers to whether the narrator can maintain and elaborate emotionally on a topic, if he/she can come to a resolution or is able to reach closure. Score 0 is given for narratives that are substantially off topic or are difficult to be defined in terms of a certain theme (possible themes could be e.g. the loss of a family member, a car accident, marriage). If the topic is identifiable, but minimally elaborated upon with personal evaluations (e.g.: I felt really sad, I was full of joy), score 1 is assigned. Score 2 is assigned when narratives are substantially developed around a theme and there are multiple personal interpretations or evaluations given. Finally, score 3 means that the narrator can take some meta-perspective on the event, and doesn't only elaborate on it with momentary feelings or evaluations, but can also link it with other autobiographical events (e.g. that reminded me of the first time I fell in love), or can come to a resolution (e.g. that event made me realize how important family is for me) or reaches closure (e.g. I feel like in the end I was able to give the event a place and move on with life). The coherent stories we created all received a score of 9, indicating they were very coherent and all incoherent stories received a score of 3, being very incoherent (score 9: Context = 3, Chronology = 3, Theme = 3; score 3: Context = 1, Chronology = 1, Theme = 1).

Subsequently, these 16 stories were evaluated on emotional valence, which means the extent to which they were positive or negative, by seven new independent raters. They received the instructions to read the stories closely and to indicate how negative or positive they consider the story to be on a scale from -5 (extremely negative) to +5 (extremely positive). Valence was taken into account secondarily since coherence is usually scored in narratives about both negative and positive events. Furthermore, social sharing of emotional events can also involve both positive and negative events. Hence, it was important to investigate whether coherence of the speaker would have an overruling effect on valence in influencing evaluations of the listener (for example: even though something positive is shared, it could still be rated to be unpleasant, because it was incoherently narrated upon).

We selected the four final narratives based on their scores for coherence and emotional valence (see S2 Appendix for full stories). We made sure to select those stories that matched as closely as possible on these scores to ensure a strict manipulation of our variables of interest (Table 1). Narrative 1 was a positive ($M$ = 4.14, $SD$ = .69) coherent (Context = 3, Chronology = 3, Theme = 3) story about graduating from high school. Narrative 2 was a positive ($M$ = 4.14, $SD$ = 1.07) incoherent (Context = 1, Chronology = 1, Theme = 1) story about falling

**Table 1. Overview of scores of narratives on coherence and valence.**

|  | Topic | Coherence rating (0–9) | Valence rating (-5 to +5) |
|---|---|---|---|
| Narrative 1 | Graduation | 9 | 4.14 |
| Narrative 2 | Falling in love | 3 | 4.14 |
| Narrative 3 | Suicide friend | 9 | -4.14 |
| Narrative 4 | Divorce parents | 3 | -4 |

in love. Narrative 3 was a negative ($M$ = -4.14, $SD$ = .90) coherent (Context = 3, Chronology = 3, Theme = 3) story about the suicide of a friend. Narrative 4 was a negative ($M$ = -4.00, $SD$ = .82) incoherent (Context = 1, Chronology = 1, Theme = 1) story about the divorce of parents.

Finally, we asked four women aged between 23 and 25 (A, B, C, D) to each voice record all four stories (1, 2, 3, 4), providing us with 16 audio clips. All stories had a word count between 295 and 304 words, resulting in a spoken duration between 90 and 105 seconds; a time that we considered reasonable for unbroken speech from one person to another. Audio instead of video clips were used to eliminate all potential visual confounders. The choice for female voices was made because of a better match between speaker and listener (87.5% female) characteristics [37].

**Questionnaires after each narrative.** We used questionnaires to investigate a variety of social evaluations with respect to the (in)coherent stories (See S3 Appendix for full questionnaires).

We measured willingness to interact with the speaker, using a questionnaire of Coyne [22]. This consisted of 8 questions, each to be answered on a 6-point Likert scale (ranging from 'absolutely not' to 'absolutely yes'), giving a possible minimum score of 8 and a maximum score of 48. Questions contained, for example, the willingness to meet the speaker, seek advice from the speaker and sit on the bus with the speaker.

We measured social support with the 2-Way Social Support Scale of Shakespeare-Finch and colleagues [38], using 3 items measuring emotional support and 2 items measuring instrumental support. Emotional support assessed elements like: I would be there to listen to his/her problems, whereas instrumental support measured things as: I would help him/her when he/she is too busy to get everything done. Both were rated on the same 6-point Likert scale (ranging from 'absolutely not' to 'absolutely yes'), resulting in a minimum score of 3 and a maximum score of 18 for emotional support and a minimum score of 2 and a maximum score of 12 for instrumental support.

We assessed momentary positive and negative feelings towards the speaker and experiencing themselves using 4 items (How much positive feelings do you have for the speaker at the moment?, How much negative feelings do you have for the speaker at the moment?, How much positive feelings do you have yourself at the moment?, How much negative feelings do you experience yourself at the moment?). Each question was to be rated on a similar 6-point Likert scale (ranging from 'absolutely not' to 'absolutely yes'), giving a minimum score of 1 and a maximum score of 6 on each of the four items.

We assessed our other variables of interest, which were trust and empathy with 9 questions, each to be answered on a similar 6-point Likert scale (ranging from 'absolutely not' to 'absolutely yes'), giving a possible minimum score of 9 and a maximum score of 54. Questions concerned, for example, how well they can empathize with the speaker, to what extent they could trust the speaker etc.

**General questionnaires.** We used some general questionnaires to look into characteristics of the participant that may have influenced the social evaluations of the narratives. We know that there are individual differences between people in their social response style, their social

bonding, and the social support they give others in general [39,40]. Furthermore there is evidence on the impact of mood (disorders) on information processing [41,42].We assessed psychological well-being, internalizing symptoms, empathy and personality characteristics to investigate whether and to what extent these variables are related to those individual differences. To measure psychological well-being, we used the Flourishing Scale (FS) [43,44]. This is a short 8-item instrument to measure psychosocial prosperity. It has good psychometric properties and is related to other psychological well-being scales [45]. To examine symptoms of depression, anxiety and stress, we used the Depression Anxiety Stress Scales (DASS-21) [46,47]. Reliability and validity of this instrument were tested and shown to be sufficient in a Dutch sample of students, which is comparable to our sample [46]. For personality characteristics, we used the Big Five Inventory (BFI) [48,49]. The Dutch BFI scales show similar psychometrics properties to the English version, namely good internal consistency and relative independence [49]. We measured empathy with our own authorized Dutch translation of the Toronto Empathy Questionnaire (TEQ) [50,51], which has been shown to be a brief, reliable and valid instrument.

## Procedure

Participants were invited to the lab in groups of maximum 6 people and were first given general information about the aim of the study. They were told that that they would be participating in a study on memory processes and aspects of social-psychological functioning. Then, the participants were asked to take place in an individual cubicle (soundproof cabinet in which they sat behind a table facing only the computer, which was connected to headphones) and to carefully read the informed consent. Herein, we stated, along with all the necessary ethical information, that we are trying to obtain insight into how individuals react when listening to memories of other people via some audio clips and behaviour- and emotion questionnaires. Upon agreement, they were informed that they would hear four different people talking about a personal memory. They were asked to pay close attention to each audio clip, as further questions about each narrative would follow. Furthermore, we told participants that after hearing the four stories and filling out the related questions, some general questionnaires would be administered. They were made aware that they could withdraw from participation at any time. Then, if the participants did not have any further questions, the headphones were put on and the experiment was initiated.

In the experiment, the specific combination of the voice and the story was counterbalanced over participants (4! = 24 possible story-voice combinations; no voice exclusively linked to a certain narrative) and administered in randomized order (computer-based randomization). Consequently, each participant heard four different stories in a random order, with each story narrated by a different voice. Each of the four audio clips was followed by questions to assess a range of the participants' social evaluations of the so-called speaker. These questions regarded willingness to interact, empathy, trust, positive and negative feelings, emotional and instrumental support. At the end of the study, participants were asked to fill out the general questionnaires. These concerned their own psychological well-being, feelings of depression, anxiety, stress, personality characteristics and trait empathy.

When participants finished the experiment, they were thanked for their participation and given a debriefing letter to take home. Herein we described the specific aim of our study, namely that we were investigating social evaluations of memory coherence, and our interest in coherence because of its relation to psychological well-being. The debriefing letter also included contact details of the researchers as well as instances for mental support, in case of any further questions or difficulties after their participation.

The main research questions, key variables, conditions and analyses were pre-registered on AsPredicted ('Narrative coherence and the response of others', #9816, https://aspredicted.org/y3d8n.pdf).

## Results

Data were analysed using repeated-measures analysis of variance (rm-ANOVA) with Coherence (Coherent, Incoherent) and Valence (Positive, Negative) as within-subjects factors, to test our hypothesis that stories that are told coherently would be more positively socially evaluated than incoherent stories. An alpha-level of .05 was set for all analyses. Follow-up paired sample

**Table 2. Descriptive statistics for social evaluations.**

| Variables | Min | Max | M | SD |
|---|---|---|---|---|
| CP/W | 8 | 48 | 34.36 | 7.76 |
| ICP/W | 12 | 48 | 30.61 | 8.13 |
| CN/W | 14 | 48 | 35.08 | 6.87 |
| ICN/W | 16 | 48 | 32.28 | 7.56 |
| CP/ES | 8 | 18 | 14.00 | 2.76 |
| ICP/ES | 3 | 18 | 13.33 | 3.30 |
| CN/ES | 6 | 18 | 14.92 | 2.75 |
| ICN/ES | 6 | 18 | 14.86 | 2.76 |
| CP/IS | 3 | 12 | 8.07 | 2.00 |
| ICP/IS | 2 | 12 | 7.49 | 2.45 |
| CN/IS | 2 | 12 | 9.05 | 2.02 |
| ICN/IS | 2 | 12 | 8.60 | 2.20 |
| CP/PFS | 1 | 6 | 4.15 | 1.16 |
| ICP/PFS | 1 | 6 | 3.86 | 1.09 |
| CN/PFS | 1 | 6 | 4.01 | 1.12 |
| ICN/PFS | 1 | 6 | 3.65 | 1.23 |
| CP/NFS | 1 | 5 | 2.05 | 1.08 |
| ICP/NFS | 1 | 5 | 2.36 | 1.23 |
| CN/NFS | 1 | 6 | 2.34 | 1.29 |
| ICN/NFS | 1 | 6 | 2.61 | 1.33 |
| CP/PFL | 1 | 6 | 4.14 | 1.25 |
| ICP/PFL | 1 | 6 | 3.91 | 1.13 |
| CN/PFL | 1 | 6 | 3.20 | 1.20 |
| ICN/PFL | 1 | 6 | 3.18 | 1.16 |
| CP/NFL | 1 | 6 | 2.13 | 1.11 |
| ICP/NFL | 1 | 5 | 2.47 | 1.12 |
| CN/NFL | 1 | 6 | 3.20 | 1.30 |
| ICN/NFL | 1 | 6 | 3.06 | 1.20 |
| CP/ET | 25 | 54 | 40.66 | 6.51 |
| ICP/ET | 19 | 52 | 35.89 | 7.52 |
| CN/ET | 20 | 54 | 40.42 | 6.87 |
| ICN/ET | 16 | 52 | 35.40 | 7.57 |

Abbreviations are Willingness (W), Emotional and Instrumental Support (ES, IS), Positive and Negative Feelings For Speaker and Listener (PFS, NFS, PFL, NFL), Empathy and Trust (ET), Coherent Positive (CP), Incoherent Positive (ICP), Coherent Negative (CN) and Incoherent Negative (ICN) narratives.

t-tests were used when rm-ANOVA results were significant ($\alpha < .05$). Analyses were executed using IBM SPSS Statistics 25.

In the analyses, we used the sum scores of the individual items for willingness (8 items), emotional support (3 items), instrumental support (2 items) and empathy and trust (9 items). Since the assessments of positive and negative feelings towards the speaker and experienced by the listener him/herself were each based on one item only, these were analysed individually. The descriptive statistics for these social evaluations can be found in Table 2.

## Main effects of memory coherence

In Fig 1, main effects of memory coherence on different social evaluations are presented. For willingness there was a significant main effect of Coherence, $F(1, 95) = 26.81$, $p < .001$, $\eta_p^2 = .22$, indicating that participants were more willing to interact with those who told a coherent story compared to those who told an incoherent story, $M_C = 34.72$, $SE_C = .65$, $M_{IC} = 31.45$, $SE_{IC} = .69$.

For instrumental support, there was a significant main effect of Coherence, $F(1, 95) = 10.96$, $p = .001$, $\eta_p^2 = .10$, as participants indicated they would give more instrumental support to someone telling a coherent story compared to someone telling an incoherent story, $M_C = 8.56$, $SE_C = .18$, $M_{IC} = 8.05$, $SE_{IC} = .21$. For emotional support however, there was not, $F(1, 95) = 2.55$, $p = .11$, $\eta_p^2 = .03$.

Participants had more positive feelings for those who told a coherent story compared to those who told an incoherent story, $M_C = 4.08$, $SE_C = .09$, $M_{IC} = 3.76$, $SE_{IC} = .09$, as indicated by a significant main effect of Coherence, $F(1, 95) = 12.35$, $p = .001$, $\eta_p^2 = .12$, on positive feelings experienced towards the speaker. Findings were similar for negative feelings experienced towards the speaker, as there was again a significant main effect of Coherence, $F(1, 95) = 9.60$, $p = .003$, $\eta_p^2 = .09$. This illustrates that participants had more negative feelings towards those who told an incoherent story compared to those who told a coherent story, $M_C = 2.20$, $SE_C = .09$, $M_{IC} = 2.49$, $SE_{IC} = .09$.

For positive feelings experienced by the listener, however, there was no main effect of Coherence, $F(1, 95) = 1.95$, $p = .17$, $\eta_p^2 = .02$. Similar results were found for negative feelings experienced by the listener, for which there was no main effect of Coherence, $F(1, 95) = 1.42$, $p = .24$, $\eta_p^2 = .02$.

With regards to trust and empathy, there was a significant main effect of Coherence, $F(1, 95) = 51.81$, $p < .001$, $\eta_p^2 = .35$. Participants trusted those who told a coherent story more compared to those who told an incoherent story, $M_C = 40.54$, $SE_C = .51$, $M_{IC} = 35.64$, $SE_{IC} = .58$.

## Interaction effects of coherence and valence

Coherence interacted with valence to impact negative feelings experienced by the listener, $F(1, 95) = 8.80$, $p = .004$, $\eta_p^2 = .09$. Follow-up paired sample t-tests showed that this interaction effect was due to a difference in responding to coherence, depending on the valence of the story. Remarkably, participants experienced more negative feelings when hearing someone telling an incoherent story compared to when hearing someone telling a coherent story, but only for stories with a positive valence, $t(95) = -2.93$, $p = .004$, not for stories with a negative valence, $t(95) = 1.12$, $p = .27$. In other words, incoherence is received more negatively when someone is talking about a positive event, $M_{CP} = 2.13$, $SE_{CP} = .11$, $M_{ICP} = 2.47$, $SE_{ICP} = .12$, whereas we feel less negative when listening to negative event narrated upon in an incoherent manner, $M_{CN} = 3.20$, $SE_{CN} = .13$, $M_{ICN} = 3.06$, $SE_{ICN} = .12$. Possible explanations for this finding will be addressed in the discussion.

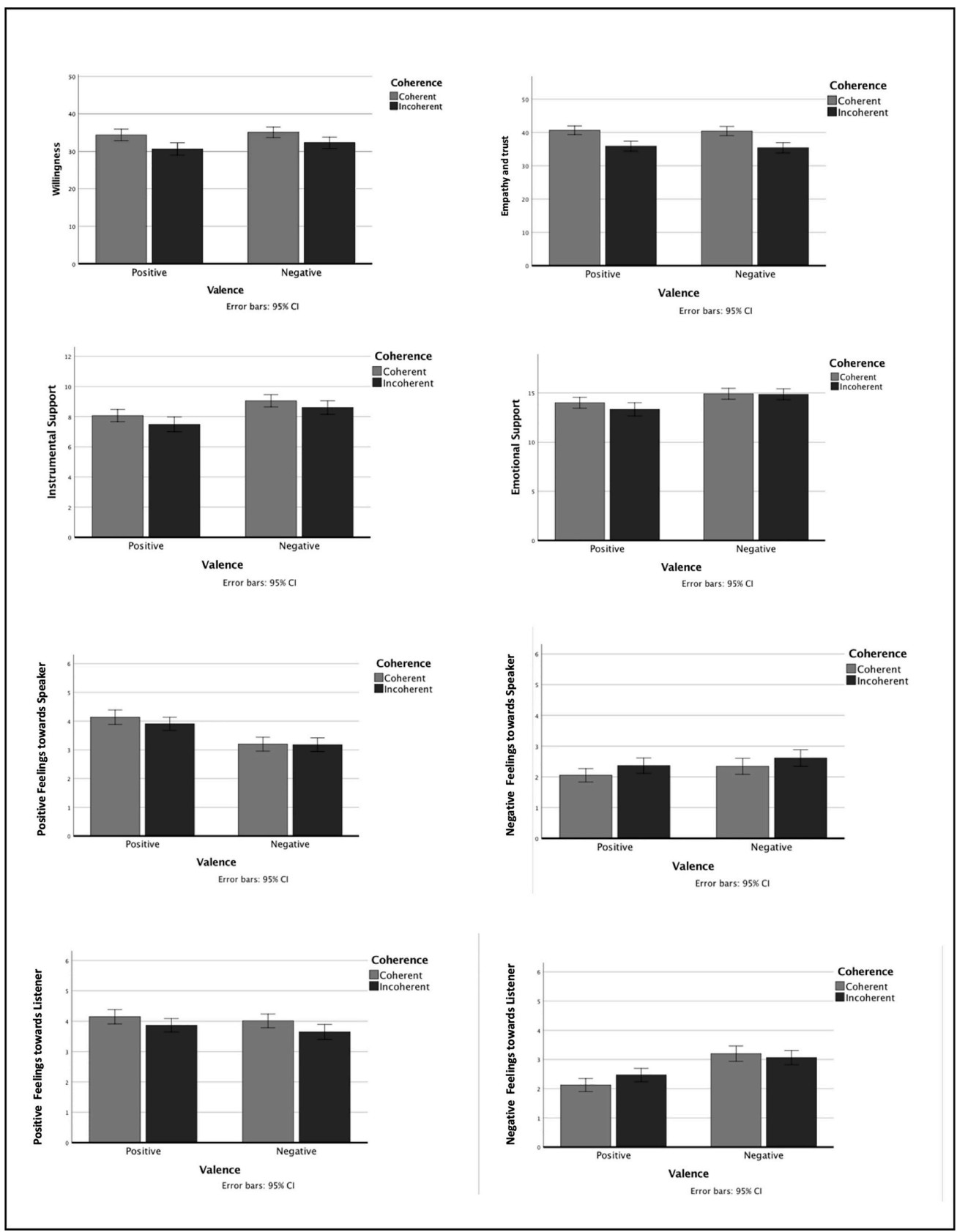

**Fig 1. Visual representation of social evaluations of autobiographical memories scores on listeners' willingness, emotional & instrumental support, positive & negative feelings towards the speaker & experienced themselves, and empathy & trust, for coherence (coherent vs incoherent) and valence (positive vs negative).**

### Individual differences in listeners

Pearson correlations between the mean scores on social evaluations over the four stories and the characteristics of the listener were calculated and Bonferroni corrections were applied ($\alpha <$ .00625) since hypotheses for this part of the analyses were exploratory. Results showed that individual differences between people in helping others, bonding, giving social support, etc. do relate to personality and mood variables. Only the results that remained significant after an appropriately conservative Bonferroni correction are discussed. Higher scores on psychological well-being were related to having more positive feelings experiencing towards the speaker, $r = .34$, $p = .001$, experiencing themselves, $r = .44$, $p < .001$, and to emotional support, $r = .29$, $p = .005$. Empathy was also positively associated with the mean of emotional support given, $r = .29$, $p = .004$. Furthermore, higher levels of neuroticism seemed to be related to a higher amount of negative feelings towards the speaker, $r = .34$, $p = .001$ and experiencing themselves, $r = .29$, $p = .004$. Lastly, depression scores were inversely related to positive feelings experiencing themselves, $r = -.43$, $p < .001$.

## Discussion

The objective of this study was to investigate the impact of autobiographical memory coherence of the speaker on social evaluations of the listener. We examined 96 participants' social evaluations of memories that were narrated in either a coherent or an incoherent manner. The results were largely in line with our hypotheses. Listeners evaluated individuals who talked about their memories in a coherent manner significantly more positively, as opposed to individuals who talked in an incoherent manner.

Two candidate mechanisms are proposed. First, as is known from the dual processing model of person cognition, impressions are formed in two stages [52,53]. The switch from the first automatic processing stage of social information to a second more controlled stage, requires increased attention, as the resource of information becomes more bottom-up rather than top-down driven [53]. Attentional effort, or effortful processing, can reduce the attraction effect [54], as social cognition and social affect do not operate independently [55]. Applied to this study, it is suggested that narrating in an incoherent fashion could require the listener to switch from an automatic mode to a controlled mode of processing social cues, increasing the attention necessary to be able to comprehend the narrative. In other words, the increased allocation of cognitive resources to understand an incoherent narrative, can generate more aversive feelings, hence increasing the likelihood of negative social evaluations. A second possible explanation comes from the idea that coherence has been seen as a "necessary but not sufficient feature of a high-quality narrative" (p. 425) [24] and "the fundamental story criterion" (p. 1193) [56]. Research in semiotic psychology supports the idea that coherence is necessary to establish truthfulness in communication [57], suggesting that listeners may perceive incoherent narratives as less truthful. Deception, even if undiscovered, has damaging effects on relationships, resulting in a mistrustful listener who is more inclined to a negative perception of the speaker [58,59]. This is in line with Conway's work, who categorized memories that score low on internal coherence as well as low on external correspondence (i.e. to reality) in the group of confabulated or false memories [60,61]. Naturally, these post-hoc explanations require further investigation making use of experimental designs.

Moreover, we also found an interaction effect of coherence and valence with regards to negative feelings that participants experienced. Participants felt more negative when listening to someone telling an incoherent story, as compared to when listening to someone telling a coherent story, but only if the stories concerned a positive theme. Incoherence in negative memories could be interpreted as a part of processing and making meaning of the event. This idea is also evident from the literature on the quality of traumatic memories, as the presence of strong negative elements can reduce the coherence with which an event is remembered, in extremer cases (traumatic memories) due to dissociative reactions [62]. However, incoherence in positive stories is not interpreted in this suggested way but reacted upon more negatively, especially because we expect positive stories to convey a pleasant message and we expect them to be entertaining. This is in compliance with evidence that positive autobiographical stories are more likely to be used for social bonding purposes, as they increase liking and interpersonal closeness more so than negative stories do [63–65].

However, this only interaction effect does not overrule all the previously discussed main effects of coherence both for positive and negative stories, so we can conclude that there is a first indication that coherence is generally socially reacted upon in a significantly more positive way than incoherence, independent of the valence of the memory. Without aiming to overinterpret these results, we do think it is important to situate the findings in the clinical field. Our results are in keeping with the idea of social maintenance of psychopathology developed by Coyne, as in his study [22] depressed individuals were characterized by a certain narrative style, which in turn accounted for negative social reactions that worsened depressive symptoms, closing the social vicious circle of mental health. In our study, narrative coherence was mainly investigated as a cause of decreased social support and thereby potentially decreased well-being, however, we adhere to a broader bidirectional perspective. The majority of research on local narrative coherence has either been correlational in nature [e.g. 27,51,52], or suggests that coherence is a consequence or symptom of different forms of psychopathology [66–68], for instance mediated by working memory load [69–71], avoidance [72] or cognitive impairment [73]. Further research is nonetheless needed for a more complete integration of social and clinical perspectives.

Since this was one of the first studies to investigate social evaluations of autobiographical memory coherence, some limitations of the current study can be taken into account in further research. These mainly relate to the ecological validity of our study, as the experimental logic gives the advantages of having more control over the effect under investigation, raising internal validity, but inevitably narrows down the complexity and overlooks the context of the phenomena investigated, lowering external validity. Alea and Bluck [37] state in their conceptual model of the social function of autobiographical memory that the memory sharing process and the specific function it serves is influenced by both speakers' and listeners' characteristics as well as their interaction. This idea is prominent in the narrative literature, for instance in research on the nature of the social relationship in which the memory sharing occurs are (e.g. peers, family members, romantic partners [5,8]), the level of responsiveness during the memory-sharing process (e.g. attentive, empathetic listening [74,75]), and the multifaceted nature of narratives (e.g. biographical embedding vs absence of it, inclusion of others' subjective perspectives vs exclusion, the extent to which the event concerned the interest of the individual and whether the participants had been through a similar situation themselves [76,77]). These contextual elements were largely kept constant, given the use of audio clips instead of real-life social interaction. Since a broad socio-cultural perspective went beyond the scope of this first study, it would be very interesting to consider these variables and their possible covariance with memory coherence in future research.

Two elements are nonetheless worth further discussing in this regard, which are gender and culture. With regards to gender, we opted to use all female voices, because we expected the main population of participants to be female. This decision was made in line with research on gender differences in narrative style [37, 78–80], making sure that the speakers' narrative style was more similar to the participants' style [81]. In line with the out-group homogeneity principle, we are more likely to form differentiated data-driven representations of individuals who are similar to ourselves (in-group, here females) than of those who are distinctly different (out-group, here males) [82].This would enhance the differentiation between speakers and result in a more nuanced social evaluation [83]. However, the fact that our sample comprised mostly young females does limit the generalizability of our results. From an early stage, research on narrative coherence has taken a sociocultural developmental perspective, focussing on how coherence comes to arise by mother-child reminiscing within a specific social and cultural context [2,7,78]. It has been shown that gender differences in autobiographical memory skills can be attributed to these conversations with parents as well as the broader interpersonal socialization [80,84]. For example, females have more detailed, vivid, emotional and longer autobiographical memories than males do, differences that are thought to be caused by differences in parent-child reminiscing [85].

Parallel to research on gender differences, early research on autobiographical memory has been characterized by the investigation of cultural differences [6,78]. The sample in this study was very homogeneous in background, as all of the participants were Belgian. Since this was only the first study in this domain, we are not able to rule out any culture-driven effects. The concept of coherence has been tested worldwide, theorized by the idea that Western and Eastern cultures differ along a dimension of individualism-collectivism in autobiographical memory [78]. Indeed, already from an early age, children from individualistic societies are found to tell more coherent, elaborated, emotional and detailed stories about their past, than children from collectivistic cultures do [86,87]. However, both cultural differences as well as similarities have been observed [88]. For example, in more recent work of Reese and colleagues [89] adolescents from three different cultural groups in New Zealand, being Māori, Chinese, and European, showed comparable age-related increases in thematic coherence over the course of development, however only European adolescents demonstrated expected age-related increases in causal coherence. Using gender diverse and multi-cultural samples in experimental studies would be an interesting route to explore in future research.

## Conclusion

Concluding, in our experimental study, listeners showed more willingness to interact, more instrumental support, more positive feelings, more empathy and more trust, towards speakers that narrated in a coherent manner about their autobiographical memories in comparison to towards those that narrated in an incoherent manner. Negative feelings in the listener were evoked when the speaker talked incoherently, but especially when it concerned a positive memory. Results can be explained in terms of a reduction in the attraction effect when effortful processing is increased, which is in line with the dual processing theory of impression formation. Another explanation involves the idea that coherence is necessary to establish truthfulness in communication. Given the sociocultural developmental pathway of narrative coherence, limitations are discussed in terms of social context, gender and cultural differences. The clinical importance of these findings is illustrated in light of the need for human connection and an interpersonal model of mental health.

## Supporting information

**S1 Appendix. Scoring criteria for the narrative coherence coding scheme (Reese et al., 2011) [24].**
(DOCX)

**S2 Appendix. Narratives (Translated from Dutch).**
(DOCX)

**S3 Appendix. Full questionnaires after each narrative (Translated from Dutch).**
(DOCX)

## Acknowledgments

The authors would like to thank the reviewers for their useful suggestions.

## Author Contributions

**Conceptualization:** Lauranne Vanaken, Dirk Hermans.

**Data curation:** Lauranne Vanaken.

**Formal analysis:** Lauranne Vanaken.

**Funding acquisition:** Dirk Hermans.

**Investigation:** Lauranne Vanaken.

**Methodology:** Lauranne Vanaken.

**Supervision:** Patricia Bijttebier, Dirk Hermans.

**Writing – original draft:** Lauranne Vanaken.

**Writing – review & editing:** Lauranne Vanaken, Dirk Hermans.

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
