## [Decision Letter · Decision Letter 0]

21 Feb 2020

PONE-D-19-29540

I like you better when you are coherent. Sharing autobiographical memories in a coherent manner has a positive impact on social responses of the listener.

PLOS ONE

Dear Vanaken,

Thank you for submitting your manuscript to PLOS ONE. After careful consideration, we feel that it has merit but does not fully meet PLOS ONE’s publication criteria as it currently stands. Therefore, we invite you to submit a revised version of the manuscript that addresses the points raised during the review process.

Both reviewers have presented main important aspects that require revisions and concern the methodology and, moreover, the overall narrative and discussion of the findings of this study. 

We would appreciate receiving your revised manuscript by Apr 06 2020 11:59PM. To enhance the reproducibility of your results, we recommend that if applicable you deposit your laboratory protocols in protocols.io, where a protocol can be assigned its own identifier (DOI) such that it can be cited independently in the future. For instructions see: http://journals.plos.org/plosone/s/submission-guidelines#loc-laboratory-protocols

We look forward to receiving your revised manuscript.

Kind regards,

Sara Rubinelli

Academic Editor

PLOS ONE

Journal Requirements:

2. We note you have included a table to which you do not refer in the text of your manuscript. Please ensure that you refer to Table 1 in your text; if accepted, production will need this reference to link the reader to the Table.

Reviewers' comments:

Reviewer's Responses to Questions

**Comments to the Author**

1. Is the manuscript technically sound, and do the data support the conclusions?

Reviewer #1: Yes

Reviewer #2: Yes

2. Has the statistical analysis been performed appropriately and rigorously? 

Reviewer #1: I Don't Know

Reviewer #2: Yes

3. Have the authors made all data underlying the findings in their manuscript fully available?

Reviewer #1: Yes

Reviewer #2: Yes

4. Is the manuscript presented in an intelligible fashion and written in standard English?

Reviewer #1: Yes

Reviewer #2: Yes

5. Review Comments to the Author

Reviewer #1: General comment:

The manuscript presents the results of an experiment aiming at investigating the effects of autobiographical memory coherence on a range of social responses of the listener. The hypothesis was that listeners would show more positive social responses (e.g. in terms of willingness to interact with the person, offering instrumental support, trust, empathy) to those people telling their story in a more coherent manner. This is relevant as these positive responses fulfill our need to belong and, by providing social support, can enhance well-being and be a protective factor for psychopathology. Despite the interest of the topic, I believe that the authors could improve their discussion section in making clearer hoe these results could be used and in which context. See below some more detailed comments.

Overall, the manuscript is well written and scientifically sound. The methods are described in sufficient detail.

Detailed comments:

-Abstract:

oalthough not required by the journal, I believe that a structured abstract is easier to read. To be reader friendly, I would suggest to add subheadings.

oplease, clearly state your objective in the abstract.

-Introduction, ll. 51-52: “This is in line with extensive evidence that a stable social network is very valuable for ensuring physical and mental health (9,10). � Please, clarify how this is related to the previous sentence.

-Introduction and Discussion: I believe that the cultural aspect should also be taken into account when discussing the value of coherence. Culture plays an important role in social relationships: the way in which people interact, what is considered appropriate or not, etc. is strongly marked by the culture. Where was the concept of coherence developed and where has it been tested until now? I wonder if everywhere in the world these three dimensions are so fundamental for the receiver.

-Discussion: In the limitations of the study a further aspect needs to be acknowledged: the fact that the study is conducted with a sample of Dutch participants. Considering that this is one of the first studies examining the responses from listeners, we cannot exclude that these results are influenced by the common cultural background of the participants.

-Discussion: With regard to the training suggested a the end of the manuscript, I believe that the authors could do an effort in making their suggestion a bit more concrete. Before introducing a training, the authors should explain how to identify the persons who tend to present their memories lacking coherence. Furthermore, they could suggest who could deliver this training, in which context (e.g. a therapy or counselling session?), etc.

-Discussion: I found the paragraph on the relationship with a therapist a bit confusing. In the rest of the manuscript the focus is on general interpersonal relationships. If the authors want to draw conclusions on the relationship with a healthcare professional, it would be appropriate to also refer to literature in the field.

-Discussion: Do the authors think that this work has clinical relevance? If yes, I think it would be important to highlight it.

-Discussion ll.395-401: Here the results are summarized but not sufficiently discussed and put into context. What can we learn from these results? What is new? What are the implications of these results?

-Discussion: ll. 373 e ff: “This means that listeners did show more positive social responses to those people who talk about their memories in a coherent manner as opposed to those who talk in an incoherent manner.” � Please, revise this sentence. It seems that a preposition is missing.

-Conclusion: Currently a conclusion is lacking. Please, summarize in a paragraph your main message for the readership.

Reviewer #2: This paper reports on an interesting study of auditory narrative coherence of positive and negative autobiographical events and its impact on the self-reported social evaluations of the listener. Its strength is in finding a negative social evaluation of an incoherent speaker, especially in the case of positive events. The paper is well written overall, but there are a few places for improvements, clarification and supporting references. I will provide a few main comments, followed by minor comments.

1. There is scope for further developing the discussion. Overall, the highlighting of the key results felt a little repetitive and interpretation of the findings needed more thinking through. More effort could be made to integrate existing literature, and care when considering speculative interpretations of the data. It is mentioned on p19 that '..could be due to the fact we feel more tolerant towards people who went through something negative!. There's no supporting references here. First, I'm not sure we can state that this is fact and therefore rephrase. Second, is there any supporting evidence for this interpretation? If not, what leads you to suggest this is higher tolerance, as opposed to reflecting empathy towards the listener or an underlying attribution of the cause of the speaker's incoherence? The listener actively makes meaning from non-verbal cues such as incoherence in speech, and inconsistencies such as incoherence in recalling a happy event. People may hold an unconscious association between incoherence and how confident we are about the truthfulness of the narrative, which is not skewed by an individual's mental state. This study itself suggests that the listener's evaluation was impacted by their own mental state, which they may project into the speaker. Needless to say, we feel more positive about listening to events we believe were as the speaker says they were. We may also implicitly know that negative events impede on how well we report it and may speculate that there's something more underlying it. Intention is also imbued by the listener. My point really is about investigating the possible interpretations further and scratching below the surface to consider why people are more negative about incoherent narratives especially happy events.

2. Mental health is mentioned in both the introduction and the clinical implications, in relation to the social function of sharing autobiographical memories. I do believe there is some relevance here in terms of the function of sharing event memories. However, this is complicated by an understanding of how mental health itself affects autobiographical memory recollection and speech coherence. This is most studied in psychosis samples and in the Adult Attachment Interview in which speech coherence (including the quality of the speech itself, I'm not sure you look at this in your measure?) is meaningful in the context of the developing sense of self, identity relevance and social relationships. Depression and anxiety also affect speech. Therefore, the mental health implications seem somewhat overplayed in this paper, especially in the discussion. While you recognise the 'vicious cycle' aspect (and I agree with that), it seems more established that mental health affects narrative coherence rather than the other way around. I would avoid reaching too far in terms of the clinical implications. The findings have a range of implications for social psychology and persuasive communication / public speaking. You can think creatively about this.

3. Throughout, the writing could be more concise in places. In particular, I sense the results could be presented in a way that would be easier for the reader to follow. Table 2 is very hard to unpick and looks like something that should be in a thesis appendix rather than in research paper. A more useful table would summarise the data in such a way that would showcase the results you report and would not require all these acronyms that are hard to make sense of. Could a table include the descriptives in the text? Also, on p16, the first sentence of paragraph 3 and paragraph 4 seem to be saying the same thing or is there a typo? They are inconsistent to each other, or have I missed something? Alternatively, could p values (and means/SDs?) appear in the figure? Overall, the text could be reduced by organising them within table or figure format. Also, in the section 'Individual differences in listener', the first part could go in an analysis section or integrated elsewhere earlier in the paper, and focus on reporting the specific results. This could be presented as a table. Make sure that the questionnaires reported in your measures are consistent with what you report, as I'm not sure I see any personality factors here.

Minor comments

1. A few places throughout would benefit from supporting citations. e.g. Introduction line 5 after 'social bonding'; materials and measures p7 - after 'hundreds of narratives in similar samples' - example refs, please.

2. Take care with you wording in places. I recognise that English may not be the first language here, so I would suggest that the term, audio 'fragments' be replaced by 'clips' or 'segments', as the term 'fragments' connotes part-words or otherwise a very small piece of narrative. This could be more semantically related, but I would also say that the participants' questionnaire responses are not 'social responses' or 'reactions' as mentioned throughout, because you did not measure a social response, which suggests observational or otherwise rated data. Rather, it's self-reported social evaluations. Social desirability effects are potentially strong in this study. Other minor wording: p10 under Table 1, 'Finally we asked four women AGED'. Also, I think audio not video was used to eliminate all potential visual confounders. Also, research 'suggests', not 'states'. Avoid using 'proven' in this kind of research. Do you mean' depression scores' rather than 'depression' (suggesting a category of people)?

3. I find the description of the participants section rather vague in terms of the reporting of sample characteristics. Can we have specific percentages/numbers?

4. On p8, the description of the coding system would really hugely benefit from being put in table format to help the reader quick reference.

5. p11: Does 'the other' refer to the speaker of the narrative? I find this wording confusing. Please rephrase. I'm not sure what is meant by 'feelings toward the other and experiencing themselves' on p11, third new paragraph.

6. I'd like to see at the beginning of the result section, what the level of agreement was in results between the different stories? Given that these stories were created specifically for this study, this would help lend some face validity to the process, as these stories haven't previously been validated.

On p18, you state 'dependent variables', but these are not DV's in the correlational analysis. Please reword.

7. The limitations section is usually dealt with in 1 paragraph in a paper. It could be more focused by not including points that are not directly about the weaknesses of the study.

8. p21, Third para: I'm not sure what you mean by 'carrying implications that cannot reduced to social interaction alone' - clarify, please.

9. It's unclear how the women who voice recorded the four stories were instructed to provide coherent and incoherent stories. It's worth noting that you controlled (roughly) length of speech but many would consider lack of coherence to be overly long narratives (and very abrupt ones). I'm also unclear why you wanted a match between the gender of the speaker and the listener. Were there any gender differences (I realise the sample is mainly female, but I'd like to rule this out).

10. It's unclear whether any valence effects are due to the narrative content or the valence itself. People are generally uncomfortable listening to a stranger talk about a suicidal (not suicide, grammar) friend and divorce(d) parents, so could this have affected listener evaluations and possibly masked some effects?

Overall, an interesting paper with potential, but some parts are quite heavy going (e.g. tables, or a lack of table) and more development is needed to give a well rounded interpretation of the findings. The reader needs help in making sense of the sheer number of variables involved.

6. PLOS authors have the option to publish the peer review history of their article (what does this mean?). If published, this will include your full peer review and any attached files.

Reviewer #1: No

Reviewer #2: No

---

## [Author Response · Author response to Decision Letter 0]

27 Feb 2020

Response to Reviewers

In what follows, we provide detailed answers to the reviewers’ comments (R: Replies) and adaptations made to the original text (M: Manuscript).

Reviewer #1: 

- C1: General comment:

The manuscript presents the results of an experiment aiming at investigating the effects of autobiographical memory coherence on a range of social responses of the listener. The hypothesis was that listeners would show more positive social responses (e.g. in terms of willingness to interact with the person, offering instrumental support, trust, empathy) to those people telling their story in a more coherent manner. This is relevant as these positive responses fulfill our need to belong and, by providing social support, can enhance well-being and be a protective factor for psychopathology. Despite the interest of the topic, I believe that the authors could improve their discussion section in making clearer hoe these results could be used and in which context. See below some more detailed comments.

Overall, the manuscript is well written and scientifically sound. The methods are described in sufficient detail.

R1: Thank you for reviewing our manuscript and for considering this topic to be relevant. We adhered to your request for a more detailed discussion section, as indicated by our replies and manuscript excerpts below. 

Detailed comments:

- C2: Abstract:

although not required by the journal, I believe that a structured abstract is easier to read. To be reader friendly, I would suggest to add subheadings.

please, clearly state your objective in the abstract.

R2: Thank you for your comment. We added subheadings and also re-wrote the majority of the abstract, to have a clearer view on the context, methods, findings and possible explanations/implications. The objective is now also explicitly stated in the abstract. 

M2: Abstract

Introduction: We all have stories to tell. The stories that prevail in our conversations frequently concern significant past personal experiences and are accordingly based on autobiographical memory retrieval and sharing. This is in line with the social function of autobiographical memory, which embodies the idea that we share memories with others to develop and maintain social relationships. However, the successful fulfilment of this social function is dependent on phenomenological properties of the memory, which are highly inter-individually different. One important individual difference is memory coherence, operationalized as narrative coherence. The objective of this study was to investigate the impact of memory coherence on the social evaluations of listeners. We hypothesized that being incoherent in the sharing of autobiographical memories, would evoke more negative social evaluations from listeners, in comparison to coherently sharing an autobiographical memory. 

Methods: In a within-subject experimental study, 96 participants listened to four pre-recorded audio clips in which the speaker narrated about an autobiographical experience, in either a coherent or an incoherent manner. 

Results: Results were in line with our hypotheses. Participants showed more willingness to interact, more instrumental support, more positive feelings, more empathy and more trust towards those narrators who talked in a coherent manner about their autobiographical memories, as compared to those that talked in an incoherent manner. Negative feelings in the listener were evoked when the speaker talked incoherently, but especially when it concerned a positive memory.

Discussion: Results can be explained in terms of a reduction in the attraction effect when effortful processing is increased, which is in line with the dual processing theory of impression formation. Another explanation involves the idea that coherence is necessary to establish truthfulness in communication. The clinical relevance of these findings is further illustrated in light of the relation between social support and psychological well-being. 

- C3: Introduction, ll. 51-52: “This is in line with extensive evidence that a stable social network is very valuable for ensuring physical and mental health (9,10). � Please, clarify how this is related to the previous sentence.

R3: We agree that the transition between those sentences was insufficient. We added a sentence and reworded a couple of adherent phrases to have a smoother text flow.

M3: Besides frequent use, the social sharing of memories also has an important function. It facilitates the development and maintenance of social relationships over time and thereby fulfils our primary need to belong (7). For example, Alea and Bluck (8) found that indicators of intimacy, such as warmth and closeness to others, increased after narrating about personally experienced relationship events (e.g. their own vacation) as opposed to talking about non-autobiographical vignettes (e.g. another couple’s vacation). The fulfilment of our need to belong or our need for human connection, has major implications for our psychological well-being, as is extensively evidenced by literature indicating that a stable social network is essential for ensuring physical and mental health (9,10). It has been repeatedly shown that good social support enhances resilience to stress and is a protective factor for psychopathology (11). Likewise, a lack of social support has been associated with a decrease in psychological well-being and a higher likelihood of developing feelings of loneliness, symptoms of depression (12,13). In sum, sharing personal memories allows us to develop a social network, thereby providing us with a sense of belonging, which is vital for our psychological well-being.

-C4: Introduction and Discussion: I believe that the cultural aspect should also be taken into account when discussing the value of coherence. Culture plays an important role in social relationships: the way in which people interact, what is considered appropriate or not, etc. is strongly marked by the culture. Where was the concept of coherence developed and where has it been tested until now? I wonder if everywhere in the world these three dimensions are so fundamental for the receiver.

R4: Thank you for your interesting question and suggestion. We agree that culture does play an important role in social relationships, and this idea is also evidenced by literature. The concept of coherence as we use it, has been defined by Reese and colleagues (2011). She has done several studies in multicultural samples, which consisted of New Zealand (NZ) Māori, Chinese, and European adolescents (Reese et al., 2014, 2017). Other research groups have also investigated cultural differences with regards to narrating life stories (e.g. Altunnar & Habermas, 2018). We have significantly elaborated on this topic in the discussion. 

M4: (…) Parallel to research on gender differences, early research on autobiographical memory has been characterized by the investigation of cultural differences (6,80). The sample in our study was very homogeneous in background, as all of the participants were Belgian. Since this was only the first study in this domain, we are not able to rule out any culture-driven effects. The concept of coherence has been tested worldwide, theorized by the idea that Western and Eastern cultures differ along a dimension of individualism-collectivism in autobiographical memory (80). Indeed, already from an early age, children from individualistic societies are found to tell more coherent, elaborated, emotional and detailed stories about their past, than children from collectivistic cultures do (88,89). However, both cultural differences as well as similarities have been observed (90). For example, in more recent work of Reese and colleagues (91) adolescents from three different cultural groups in New Zealand, being Māori, Chinese, and European, showed comparable age-related increases in thematic coherence over the course of development, however only European adolescents demonstrated expected age-related increases in causal coherence. Using gender diverse and multi-cultural samples in experimental studies would be an interesting route to explore in future research. 

- C5: Discussion: In the limitations of the study a further aspect needs to be acknowledged: the fact that the study is conducted with a sample of Dutch participants. Considering that this is one of the first studies examining the responses from listeners, we cannot exclude that these results are influenced by the common cultural background of the participants.

R5: It is indeed true that we cannot exclude any culture-specific effects. As illustrated above (M4), we have integrated your comment about the background of the participants in the paragraph on cultural differences, which we have significantly elaborated on in the discussion. 

- C6: Discussion: With regard to the training suggested a the end of the manuscript, I believe that the authors could do an effort in making their suggestion a bit more concrete. Before introducing a training, the authors should explain how to identify the persons who tend to present their memories lacking coherence. Furthermore, they could suggest who could deliver this training, in which context (e.g. a therapy or counselling session?), etc.

R6: Thank you for your comment. We left out the part on training since we think these results are still to be further developed and replicated before some sort of training could be installed. In general, we do not yet know enough about the mechanisms behind memory coherence, so future theoretical work would be needed before application possibilities can be investigated.

- C7: Discussion: I found the paragraph on the relationship with a therapist a bit confusing. In the rest of the manuscript the focus is on general interpersonal relationships. If the authors want to draw conclusions on the relationship with a healthcare professional, it would be appropriate to also refer to literature in the field.

R7: Thank you for your suggestion. We agree that this part was not really integrated with the rest of manuscript, as indeed it was specifically on a certain kind of professional relationship, as opposed to general interpersonal relationships. We decided to leave out this part as well. In R8, more explanations about possible clinical relevance is provided.

- C8: Discussion: Do the authors think that this work has clinical relevance? If yes, I think it would be important to highlight it.

R8: Clinical relevance can be perceived from the standpoint of a scientist (mechanisms of psychopathology) vs the standpoint of a clinician (tools for clinical practice). We do think, taking the standpoint of a scientist, that these findings can have social and clinical relevance, in the sense that they help explain mechanisms of psychopathology. As elaborated in the introduction, we propose that social functioning could be a mechanism in the relation between narrative coherence and psychopathology. However, with regards to direct use for clinical practice, we think further replications are needed before interventions can be developed from these findings, since this in only the first study in this domain. We want to be more careful regarding overinterpretations of findings, hence we left out the initial part on clinical applicability (training and therapeutic relationship, as indicated above: R6 & R7), and re-wrote a more nuanced point of discussion. This involves the idea of social maintenance of psychopathology as well as the broader bidirectional relation between coherence and mental health. 

M8: Without aiming to overinterpret these results, we do think it is important to situate the findings in the clinical field. Our results are in keeping with the idea of social maintenance of psychopathology developed by Coyne, as in his study (22) depressed individuals were characterized by a certain narrative style, which in turn accounted for negative social reactions that worsened depressive symptoms, closing the social vicious circle of mental health. In our study, narrative coherence was mainly investigated as a cause of decreased social support and thereby potentially decreased well-being, however, we adhere to a broader bidirectional perspective. The majority of research on local narrative coherence has either been correlational in nature (e.g. 27,51,52), or suggests that coherence is a consequence or symptom of different forms of psychopathology (68–70), for instance mediated by working memory load (71–73), avoidance (74) or cognitive impairment (75). Further research is nonetheless needed for a more complete integration of social and clinical perspectives.

- C9: Discussion ll.395-401: Here the results are summarized but not sufficiently discussed and put into context. What can we learn from these results? What is new? What are the implications of these results?

R9: Thank you for your comment. We agree that there was a great majority of the discussion spent on repeating the results, whereas interpretations and implications of findings were lacking. Hence, we considerably changed the entire discussion section. Two candidate mechanisms are proposed to explain our main effects of coherence on social evaluations. A first explanation that was elaborated upon was one in terms of a reduction in the attraction effect when effortful processing is increased, which is in line with the dual processing theory of impression formation. Another explanation involves the idea that coherence is necessary to establish truthfulness in communication. Furthermore, to explain the interaction effect of coherence and valence, meaningful links to existing literature were made, like the use of positive memories for social bonding. These comments were integrated in the manuscript as follows. 

M9: The objective of this study was to investigate the impact of autobiographical memory coherence of the speaker on social evaluations of the listener. We examined 96 participants’ social evaluations of memories that were narrated in either a coherent or an incoherent manner. The results were largely in line with our hypotheses. Listeners evaluated individuals who talked about their memories in a coherent manner significantly more positively, as opposed to individuals who talked in an incoherent manner. 

 Two candidate mechanisms are proposed. First, as is known from the dual processing model of person cognition, impressions are formed in two stages (53,54). The switch from the first automatic processing stage of social information to a second more controlled stage, requires increased attention, as the resource of information becomes more bottom-up rather than top-down driven (54). Attentional effort, or effortful processing, can reduce the attraction effect (55), as social cognition and social affect do not operate independently (56). Applied to this study, it is suggested that narrating in an incoherent fashion could require the listener to switch from an automatic mode to a controlled mode of processing social cues, increasing the attention necessary to be able to comprehend the narrative. In other words, the increased allocation of cognitive resources to understand an incoherent narrative, can generate more aversive feelings, hence increasing the likelihood of negative social evaluations. A second possible explanation comes from the idea that coherence has been seen as a ‘‘necessary but not sufficient feature of a high-quality narrative” (p. 425) (24) and ‘‘the fundamental story criterion” (p. 1193) (57). Research in semiotic psychology supports the idea that coherence is necessary to establish truthfulness in communication (58), suggesting that listeners may perceive incoherent narratives as less truthful. Deception, even if undiscovered, has damaging effects on relationships, resulting in a mistrustful listener who is more inclined to a negative perception of the speaker (59,60).This is in line with Conway’s work, who categorized memories that score low on internal coherence as well as low on external correspondence (i.e. to reality) in the group of confabulated or false memories (61,62). Naturally, these post-hoc explanations require further investigation making use of experimental designs. 

 Moreover, we also found an interaction effect of coherence and valence with regards to negative feelings that participants experienced. Participants felt more negative when listening to someone telling an incoherent story, as compared to when listening to someone telling a coherent story, but only if the stories concerned a positive theme. Incoherence in negative memories could be interpreted as a part of processing and making meaning of the event. This idea is also evident from the literature on the quality of traumatic memories, as the presence of strong negative elements can reduce the coherence with which an event is remembered, in extremer cases (traumatic memories) due to dissociative reactions (63). However, incoherence in positive stories is not interpreted in this suggested way but reacted upon more negatively, especially because we expect positive stories to convey a pleasant message and we expect them to be entertaining. This is in compliance with evidence that positive autobiographical stories are more likely to be used for social bonding purposes, as they increase liking and interpersonal closeness more so than negative stories do (64–66).

-C10: Discussion: ll. 373 e ff: “This means that listeners did show more positive social responses to those people who talk about their memories in a coherent manner as opposed to those who talk in an incoherent manner.” � Please, revise this sentence. It seems that a preposition is missing.

R10: Thanks for noticing. Since the entire discussion was changed, this sentence is now no longer in the manuscript.

-C11: Conclusion: Currently a conclusion is lacking. Please, summarize in a paragraph your main message for the readership.

R 11: We added a conclusion to the text, summarizing the main message. 

M11: Conclusion

 Concluding, in our experimental study, listeners showed more willingness to interact, more instrumental support, more positive feelings, more empathy and more trust, towards speakers that narrated in a coherent manner about their autobiographical memories in comparison to towards those that narrated in an incoherent manner. Negative feelings in the listener were evoked when the speaker talked incoherently, but especially when it concerned a positive memory. Results can be explained in terms of a reduction in the attraction effect when effortful processing is increased, which is in line with the dual processing theory of impression formation. Another explanation involves the idea that coherence is necessary to establish truthfulness in communication. Given the sociocultural developmental pathway of narrative coherence, limitations are discussed in terms of social context, gender and cultural differences. The clinical importance of these findings is illustrated in light of the need for human connection and an interpersonal model of mental health. 

We would like to thank Reviewer 1 again for reading the manuscript and providing useful comments and questions. We hoped to have provided you with all needed explanations in the replies and to have encompassed all comments in the revised version of the manuscript.  

Reviewer #2: 

This paper reports on an interesting study of auditory narrative coherence of positive and negative autobiographical events and its impact on the self-reported social evaluations of the listener. Its strength is in finding a negative social evaluation of an incoherent speaker, especially in the case of positive events. The paper is well written overall, but there are a few places for improvements, clarification and supporting references. I will provide a few main comments, followed by minor comments.

R: Thank you for reviewing our manuscript and for considering this study to be interesting.

1. There is scope for further developing the discussion. Overall, the highlighting of the key results felt a little repetitive and interpretation of the findings needed more thinking through. More effort could be made to integrate existing literature, and care when considering speculative interpretations of the data. It is mentioned on p19 that '..could be due to the fact we feel more tolerant towards people who went through something negative!. There's no supporting references here. First, I'm not sure we can state that this is fact and therefore rephrase. Second, is there any supporting evidence for this interpretation? If not, what leads you to suggest this is higher tolerance, as opposed to reflecting empathy towards the listener or an underlying attribution of the cause of the speaker's incoherence? The listener actively makes meaning from non-verbal cues such as incoherence in speech, and inconsistencies such as incoherence in recalling a happy event. People may hold an unconscious association between incoherence and how confident we are about the truthfulness of the narrative, which is not skewed by an individual's mental state. This study itself suggests that the listener's evaluation was impacted by their own mental state, which they may project into the speaker. Needless to say, we feel more positive about listening to events we believe were as the speaker says they were. We may also implicitly know that negative events impede on how well we report it and may speculate that there's something more underlying it. Intention is also imbued by the listener. My point really is about investigating the possible interpretations further and scratching below the surface to consider why people are more negative about incoherent narratives especially happy events.

R1: Thank you for your comment. We agree that there was a great majority of the discussion spent on repeating the results, whereas interpretation of findings was lacking. Hence, we considerably changed the entire discussion section. Parts that were not supported by evidence and mere intuitive interpretation (e.g. could be due to the fact we feel more tolerant towards people who went through something negative) were left out. Instead, meaningful links to existing literature were made, like the use of positive memories for social bonding. Your suggestion as to why people are more negative about incoherent events gave also rise to discussion a literature search. We suggested two candidate mechanisms for the observed effects. One concerned the reduction of the attraction effect when increased effortful processing, the other concerned the association between coherence and the truthfulness of the narrative (as you suggested). Furthermore, we do want to reiterate that it was only with regards to 1 variable (negative feelings of the listener), that coherence interacted with valence, for all other variables, coherence had an overruling effect. These comments were integrated in the manuscript as follows. 

M1: The objective of this study was to investigate the impact of autobiographical memory coherence of the speaker on social evaluations of the listener. We examined 96 participants’ social evaluations of memories that were narrated in either a coherent or an incoherent manner. The results were largely in line with our hypotheses. Listeners evaluated individuals who talked about their memories in a coherent manner significantly more positively, as opposed to individuals who talked in an incoherent manner. 

 Two candidate mechanisms are proposed. First, as is known from the dual processing model of person cognition, impressions are formed in two stages (53,54). The switch from the first automatic processing stage of social information to a second more controlled stage, requires increased attention, as the resource of information becomes more bottom-up rather than top-down driven (54). Attentional effort, or effortful processing, can reduce the attraction effect (55), as social cognition and social affect do not operate independently (56). Applied to this study, it is suggested that narrating in an incoherent fashion could require the listener to switch from an automatic mode to a controlled mode of processing social cues, increasing the attention necessary to be able to comprehend the narrative. In other words, the increased allocation of cognitive resources to understand an incoherent narrative, can generate more aversive feelings, hence increasing the likelihood of negative social evaluations. A second possible explanation comes from the idea that coherence has been seen as a ‘‘necessary but not sufficient feature of a high-quality narrative” (p. 425) (24) and ‘‘the fundamental story criterion” (p. 1193) (57). Research in semiotic psychology supports the idea that coherence is necessary to establish truthfulness in communication (58), suggesting that listeners may perceive incoherent narratives as less truthful. Deception, even if undiscovered, has damaging effects on relationships, resulting in a mistrustful listener who is more inclined to a negative perception of the speaker (59,60). This is in line with Conway’s work, who categorized memories that score low on internal coherence as well as low on external correspondence (i.e. to reality) in the group of confabulated or false memories (61,62). Naturally, these post-hoc explanations require further investigation making use of experimental designs.

 Moreover, we also found an interaction effect of coherence and valence with regards to negative feelings that participants experienced. Participants felt more negative when listening to someone telling an incoherent story, as compared to when listening to someone telling a coherent story, but only if the stories concerned a positive theme. Incoherence in negative memories could be interpreted as a part of processing and making meaning of the event. This idea is also evident from the literature on the quality of traumatic memories, as the presence of strong negative elements can reduce the coherence with which an event is remembered, in extremer cases (traumatic memories) due to dissociative reactions (63). However, incoherence in positive stories is not interpreted in this suggested way but reacted upon more negatively, especially because we expect positive stories to convey a pleasant message and we expect them to be entertaining. This is in compliance with evidence that positive autobiographical stories are more likely to be used for social bonding purposes, as they increase liking and interpersonal closeness more so than negative stories do (64–66).

2. Mental health is mentioned in both the introduction and the clinical implications, in relation to the social function of sharing autobiographical memories. I do believe there is some relevance here in terms of the function of sharing event memories. However, this is complicated by an understanding of how mental health itself affects autobiographical memory recollection and speech coherence. This is most studied in psychosis samples and in the Adult Attachment Interview in which speech coherence (including the quality of the speech itself, I'm not sure you look at this in your measure?) is meaningful in the context of the developing sense of self, identity relevance and social relationships. Depression and anxiety also affect speech. Therefore, the mental health implications seem somewhat overplayed in this paper, especially in the discussion. While you recognise the 'vicious cycle' aspect (and I agree with that), it seems more established that mental health affects narrative coherence rather than the other way around. I would avoid reaching too far in terms of the clinical implications. The findings have a range of implications for social psychology and persuasive communication /public speaking. You can think creatively about this.

R2: Thank you for your comment. We agree that the literature supports a bidirectional relation between coherence and mental health. Our measure of Reese et al. (2011) differs partly from the measure that Lysaker et al. (1992) use to measure coherence, as they respectively measure local coherence (in psychopathology and healthy controls) and speech coherence (in psychosis). However, also in the literature that follows Reese’s, the idea that psychopathology affects narrative style is supported. We included studies that evidence this idea in the discussion. 

 Moreover, clinical relevance can be perceived from the standpoint of a scientist (mechanisms of psychopathology) vs the standpoint of a clinician (tools for clinical practice). We do think, taking the standpoint of a scientist, that these findings can have social and clinical relevance, in the sense that they help explain mechanisms of psychopathology. As elaborated in the introduction, we propose that social functioning could be a mechanism in the relation between narrative coherence and psychopathology. However, with regards to direct use for clinical practice, we think further replications are needed before interventions can be developed from these findings, since this in only the first study in this domain. We want to be more careful regarding overinterpretations of findings, hence we left out the initial part on clinical applicability (training and therapeutic relationship), and re-wrote a more nuanced point of discussion. This involves the idea of social maintenance of psychopathology as well as the broader bidirectional relation between coherence and mental health. Social implications (e.g., the dual processing model of person cognition) were also further elaborated upon as indicated by M1.

M2: Without aiming to overinterpret these results, we do think it is important to situate the findings in the clinical field. Our results are in keeping with the idea of social maintenance of psychopathology developed by Coyne, as in his study (22) depressive symptoms gave rise to a certain narrative style, which in turn accounted for negative social reactions that worsened depressive symptoms, closing the social vicious circle of mental health. In our study, narrative coherence was mainly investigated as a cause of decreased social support and thereby potentially decreased well-being, however, we adhere to a broader bidirectional perspective. The majority of research on local narrative coherence has either been correlational in nature (e.g. 27,51,52), or suggests that coherence is a consequence or symptom of different forms of psychopathology (68–70), for instance mediated by working memory load (71–73), avoidance (74) or cognitive impairment (75). Further research is nonetheless needed for a more complete integration of social and clinical perspectives.

3. Throughout, the writing could be more concise in places. In particular, I sense the results could be presented in a way that would be easier for the reader to follow. Table 2 is very hard to unpick and looks like something that should be in a thesis appendix rather than in research paper. A more useful table would summarise the data in such a way that would showcase the results you report and would not require all these acronyms that are hard to make sense of. Could a table include the descriptives in the text? Also, on p16, the first sentence of paragraph 3 and paragraph 4 seem to be saying the same thing or is there a typo? They are inconsistent to each other, or have I missed something? Alternatively, could p values (and means/SDs?) appear in the figure? Overall, the text could be reduced by organising them within table or figure format. Also, in the section 'Individual differences in listener', the first part could go in an analysis section or integrated elsewhere earlier in the paper, and focus on reporting the specific results. This could be presented as a table. Make sure that the questionnaires reported in your measures are consistent with what you report, as I'm not sure I see any personality factors here.

R3a: Thank you for your concern. The descriptive results are presented in Table 2 since it is an APA-rule that descriptive statistics are presented in a table in raw format, before further inferential statistics are executed (here ANOVA). In accordance with this rule, we decided to leave the descriptives in the table. We did however change some of the acronyms in the table to the same wordings as used in the text, which makes the table easier to comprehend. 

With regards to your next question, we are sorry for the confusion that have might been due to our wording. Paragraph 3 is about the feelings towards the speaker, paragraph 4 is about the feelings the listeners experienced themselves. We changed the phrasing in these paragraphs to make it easier to understand. 

M3a: Participants had more positive feelings for those who told a coherent story compared to those who told an incoherent story, MC = 4.08, SEC = .09, MIC = 3.76, SEIC = .09, as indicated by a significant main effect of Coherence, F (1, 95) = 12.35, p = .001, �p2 = .12, on positive feelings experienced towards the speaker. Findings were similar for negative feelings experienced towards the speaker, as there was again a significant main effect of Coherence, F (1, 95) = 9.60, p = .003, �p2 = .09.

R3b:Furthermore, in the figure, we chose to only include the standard deviations by means of the error bars, since means were already all individually presented in the table (APA-norm), and p-values were presented alongside their respective executed F-tests in the text. To keep the figure uncluttered and easy to interpret, numerical values were chosen to be reported in the running text.

Thanks again for your suggestion. We moved the first part of the section on 'Individual differences in listener', to the methods, so the focus is now on reporting the specific results. We chose to not work with a table here, since these results are only a response to a secondary more exploratory question, which we do not want to draw the main focus of attention to, in order to keep a smoother text flow. 

Lastly, the questionnaires reported in our results section were all consistent with our methods section. Personality factors were measured using the BFI, as shown below.

M3b: For personality characteristics, we used the Big Five Inventory (BFI) (40,41). The Dutch BFI scales show similar psychometrics properties to the English version, namely good internal consistency and relative independence (41). (…) Furthermore, higher levels of neuroticism seemed to be related to a higher amount of negative feelings towards the speaker, r = .34, p = .001 and experiencing themselves, r = .29, p = .004.

Minor comments

1. A few places throughout would benefit from supporting citations. e.g. Introduction line 5 after 'social bonding'(1A); materials and measures p7 - after 'hundreds of narratives in similar samples'(1B) - example refs, please.

R1a: Thank you for noticing. We added supporting citations to give a clearer perspective on the existing literature. M1a: Hence, it is evident that one of the three main functions of autobiographical memory is a social function, which embodies the idea that we share memories with others to develop and maintain social relationships (3–6).

3. Bluck S, Alea N, Habermas T, Rubin DC. A tale of three functions: The self-reported uses of autobiographical memory. Soc Cogn [Internet]. 2005;23(1):91–117. Available from: http://guilfordjournals.com/doi/10.1521/soco.23.1.91.59198

4. Bluck S, Alea N. Thinking and talking about the past: Why remember? Appl Cogn Psychol [Internet]. 2009 [cited 2019 Jul 24];23(8):1089–104. Available from: www.interscience.wiley.com

5. Bluck S, Alea N. Crafting the tale: Construction of a measure to assess the functions of autobiographical remembering. Memory. 2011;19(5):470–86. 

6. Nelson K. Psychological and Social Origins of Memory. Psychol Sci. 1993;4(1):7–14.

R1b: The ‘hundreds of narratives in similar samples', is based on our own work, as we are currently writing up a paper in which we coded N = 683 narratives for coherence. However, similar work came out recently, indicating those same event types in student’s narratives. We included references to those studies as well. 

M1b: We wrote the stories, based on our extensive experience collecting and coding hundreds of narratives in similar samples of our own studies (in prep), and investigating event types in similar work (25, 26, 36).

25. Mitchell C, Reese E, Salmon K, Jose P. Narrative coherence, psychopathology, and wellbeing: Concurrent and longitudinal findings in a mid-adolescent sample. J Adolesc. 2020;79:16–25. 

26. Vanderveren E, Bijttebier P, Hermans D. Autobiographical memory coherence and specificity: Examining their reciprocal relation and their associations with internalizing symptoms and rumination. Behav Res Ther. 2019;116(February):30–5.

36. McLean KC, Breen A V., Fournier MA. Constructing the self in early, middle, and late adolescent boys: Narrative identity, individuation, and well-being. J Res Adolesc [Internet]. 2010 Mar 1 [cited 2020 Jan 28];20(1):166–87. Available from: http://doi.wiley.com/10.1111/j.1532-7795.2009.00633.x

2. Take care with you wording in places. I recognise that English may not be the first language here, so I would suggest that the term, audio 'fragments' be replaced by 'clips' or 'segments', as the term 'fragments' connotes part-words or otherwise a very small piece of narrative. This could be more semantically related, but I would also say that the participants' questionnaire responses are not 'social responses' or 'reactions' as mentioned throughout, because you did not measure a social response, which suggests observational or otherwise rated data. Rather, it's self-reported social evaluations. Social desirability effects are potentially strong in this study. Other minor wording: p10 under Table 1, 'Finally we asked four women AGED'. Also, I think audio not video was used to eliminate all potential visual confounders. Also, research 'suggests', not 'states'. Avoid using 'proven' in this kind of research. Do you mean' depression scores' rather than 'depression' (suggesting a category of people)?

R2. Thank you for these useful suggestions. Since English is indeed not the first language, all the suggestions were very welcome. Tracked changes were made throughout the manuscript. 

3. I find the description of the participants section rather vague in terms of the reporting of sample characteristics. Can we have specific percentages/numbers?

R3. Specific percentages were included.

M3: A total of 96 adults between the ages of 19 and 40 (M = 21.06, SD = 3.17) participated in the study, of which 84 (87.5%) were female and 12 (12.5%) were male.

4. On p8, the description of the coding system would really hugely benefit from being put in table format to help the reader quick reference.

R4: We have included an appendix to have a quick overview of the scoring criteria. 

M4: This coding system evaluates narratives on 3 separate dimensions (score 0-3) that are summed up to entail total memory coherence (score 0-9) (See S1 Appendix for scoring criteria, adopted from Reese et al., 2011, p. 436).

S1 Appendix. Scoring criteria for the Narrative Coherence Coding Scheme (Adopted from Reese et al., 2011, p. 436)

5. p11: Does 'the other' refer to the speaker of the narrative? I find this wording confusing. Please rephrase. I'm not sure what is meant by 'feelings toward the other and experiencing themselves' on p11, third new paragraph.

R5: The other does indeed refer to the speaker of the narrative, we are sorry to make this confusing. “The other” was left out in the entire manuscript and replaced by the speaker to make things clearer. “Feelings toward the other” are the feelings they experience towards the speaker, how positive/negative they evaluate the speaker (attitude towards speaker). “Feelings experienced themselves” refers to their own momentary feelings, how positive/negative they are feeling (mood of their own). 

6. I'd like to see at the beginning of the result section, what the level of agreement was in results between the different stories? Given that these stories were created specifically for this study, this would help lend some face validity to the process, as these stories haven't previously been validated.

On p18, you state 'dependent variables', but these are not DV's in the correlational analysis. Please reword.

R6: Thank you for your correction. We changed the sentence. However, with regards to your question about the level of agreement, it is specifically the difference in responding towards a coherent vs an incoherent story that was the focus of the study. Looking at associations between social evaluations for stories differing in coherence would not answer that question and could likely distract the focus of the results section. Hence, we opted to leave this part out. Furthermore, the level of agreement between stories would not allow us to assess the validity of our experimental manipulation. But thank you for your suggestion nonetheless. 

M6: Pearson correlations between the mean scores on social evaluations over the four stories and the characteristics of the listener were calculated.

7. The limitations section is usually dealt with in 1 paragraph in a paper. It could be more focused by not including points that are not directly about the weaknesses of the study.

R7: Thank you for your suggestion. We significantly changed the discussion section. The limitations are now dealt with in 1 paragraph. We chose to include a separate paragraph on gender, in line with your suggestion below.

M7: Since this was one of the first studies to investigate social evaluations of autobiographical memory coherence, some limitations of the current study can be taken into account in further research. These mainly relate to the ecological validity of our study, as the experimental logic gives the advantages of having more control over the effect under investigation, raising internal validity, but inevitably narrows down the complexity and overlooks the context of the phenomena investigated, lowering external validity. Alea and Bluck (37) state in their conceptual model of the social function of autobiographical memory that the memory sharing process and the specific function it serves is influenced by both speakers’ and listeners’ characteristics as well as their interaction. This idea is prominent in the narrative literature, for instance in research on the nature of the social relationship in which the memory sharing occurs are (e.g. peers, family members, romantic partners (5,8)), the level of responsiveness during the memory-sharing process (e.g. attentive, empathetic listening (75,76)), and the multifaceted nature of narratives (e.g. biographical embedding vs absence of it, inclusion of others’ subjective perspectives vs exclusion, the extent to which the event concerned the interest of the individual and whether the participants had been through a similar situation themselves (77,78)). These contextual elements were largely kept constant, given the use of audio clips instead of real-life social interaction. Since a broad socio-cultural perspective went beyond the scope of this first study, it would be very interesting to consider these variables and their possible covariance with memory coherence in future research. 

8. p21, Third para: I'm not sure what you mean by 'carrying implications that cannot reduced to social interaction alone' - clarify, please.

R8: We meant that social interaction affects psychological well-being in its turn, so that narrative coherence indirectly has an effect on our mental health, via social support. However, this entire paragraph was adapted (See M2).

9. It's unclear how the women who voice recorded the four stories were instructed to provide coherent and incoherent stories. It's worth noting that you controlled (roughly) length of speech but many would consider lack of coherence to be overly long narratives (and very abrupt ones). I'm also unclear why you wanted a match between the gender of the speaker and the listener. Were there any gender differences (I realise the sample is mainly female, but I'd like to rule this out).

R9: We wrote the four stories exactly as they were to be voice recorded, (in)coherence was thus already in the written story. The specific wordings of the stories can be found in S2 Appendix. This way, we could ensure a strict manipulation of our variable of interest (coherence). 

The choice for female voices was made because research has shown that it is easier to empathize and connect with people who are more similar to ourselves (Alea & Bluck, 2007). Hence, having a female individual talk about their memories, simulated the reality of listening to a female friend better (as females are more likely to have female friends as opposed to male friends). The topic of gender was significantly extended in the discussion. 

M9: With regards to gender, we opted to use all female voices, because we expected the main population of participants to be female. This decision was made in line with research on gender differences in narrative style (37,79–81), making sure that the speakers’ narrative style was more similar to the participants’ style (82). In line with the out-group homogeneity principle, we are more likely to form differentiated data-driven representations of individuals who are similar to ourselves (in-group, here females) than of those who are distinctly different (out-group, here males) (83).This would enhance the differentiation between speakers and result in a more nuanced social evaluation (84). However, the fact that our sample comprised mostly young females does limit the generalizability of our results. From an early stage, research on narrative coherence has taken a sociocultural developmental perspective, focussing on how coherence comes to arise by mother-child reminiscing within a specific social and cultural context (2,7,79). It has been shown that gender differences in autobiographical memory skills can be attributed to these conversations with parents as well as the broader interpersonal socialization (81,85). For example, females have more detailed, vivid, emotional and longer autobiographical memories than males do, differences that are thought to be caused by differences in parent-child reminiscing (86). 

10. It's unclear whether any valence effects are due to the narrative content or the valence itself. People are generally uncomfortable listening to a stranger talk about a suicidal (not suicide, grammar) friend and divorce(d) parents, so could this have affected listener evaluations and possibly masked some effects?

R10: This is an interesting suggestion. Intuitively, it could indeed be true that people are general uncomfortable listening to a stranger talk about intense negative events. However, our results indicate that this was not the case in our study. Participants’ social evaluations of negative stories were not any more negative than those of positive stories. In our study, coherence had an overruling effect. Incoherence was negatively socially evaluated, regardless of the valence of the story. We did not include the main effects of valence in our study, since it was only assessed secondarily and we wanted to keep a focus on coherence throughout the manuscript. We did nonetheless include interaction effects of coherence and valence. Only for the variable ‘negative feelings’, there was an interaction effect between coherence and valence, indicating that listeners experienced more negative feelings when listening to an incoherent story (as opposed to coherent), but especially when it concerned stories of a positive valence. This is thus opposite to what the suggested idea would predict. 

M10: Coherence interacted with valence to impact negative feelings experienced by the listener, F (1, 95) = 8.80, p = .004, �p2 = .09. Follow-up paired sample t-tests showed that this interaction effect was due to a difference in responding to coherence, depending on the valence of the story. Remarkably, participants experienced more negative feelings when hearing someone telling an incoherent story compared to when hearing someone telling a coherent story, but only for stories with a positive valence, t(95) = -2.93, p = .004, not for stories with a negative valence, t(95) = 1.12, p = .27. In other words, incoherence is received more negatively when someone is talking about a positive event, MCP = 2.13, SECP = .11, MICP = 2.47, SEICP = .12, whereas we feel less negative when listening to negative event narrated upon in an incoherent manner, MCN = 3.20, SECN = .13, MICN = 3.06, SEICN = .12. Possible explanations for this finding will be addressed in the discussion.

Overall, an interesting paper with potential, but some parts are quite heavy going (e.g. tables, or a lack of table) and more development is needed to give a well rounded interpretation of the findings. The reader needs help in making sense of the sheer number of variables involved.

Thank you. We hope to have provided you with extensive clarifications and interpretations of the findings. 

We would like to thank Reviewer 2 again for reading the manuscript and providing useful comments and questions. We hoped to have provided you with all needed explanations in the replies and to have encompassed all comments in the revised version of the manuscript.

---

## [Decision Letter · Decision Letter 1]

10 Apr 2020

I like you better when you are coherent.

Narrating autobiographical memories in a coherent manner has a positive impact on listeners’ social evaluations.

PONE-D-19-29540R1

Dear Dr. Vanaken,

We are pleased to inform you that your manuscript has been judged scientifically suitable for publication and will be formally accepted for publication once it complies with all outstanding technical requirements.

With kind regards,

Sara Rubinelli

Academic Editor

PLOS ONE

Additional Editor Comments (optional):

Reviewers' comments:

Reviewer's Responses to Questions

**Comments to the Author**

1. If the authors have adequately addressed your comments raised in a previous round of review and you feel that this manuscript is now acceptable for publication, you may indicate that here to bypass the “Comments to the Author” section, enter your conflict of interest statement in the “Confidential to Editor” section, and submit your "Accept" recommendation.

Reviewer #1: All comments have been addressed

Reviewer #2: All comments have been addressed

2. Is the manuscript technically sound, and do the data support the conclusions?

Reviewer #1: Yes

Reviewer #2: Yes

3. Has the statistical analysis been performed appropriately and rigorously? 

Reviewer #1: I Don't Know

Reviewer #2: Yes

4. Have the authors made all data underlying the findings in their manuscript fully available?

Reviewer #1: Yes

Reviewer #2: Yes

5. Is the manuscript presented in an intelligible fashion and written in standard English?

Reviewer #1: Yes

Reviewer #2: Yes

6. Review Comments to the Author

Reviewer #1: The authors addressed all commments and made a significant effort in interpreting their results and reflecting on their implications. The manuscript has substantially improved.

Very minor points:

- in the Abstract, under the Methods, please give some indications with regards to outcomes and outcome measures.

- S1 Appendix. Scoring criteria for the Narrative Coherence Coding Scheme. Formatting/layout problem: The table in the word document looks "squeezed" and is unreadable.

- Figure 1. Formatting/layout problem: Not sharp.

Reviewer #2: The discussion is much improved with reasonable interpretation of the findings. I still find Table 2 to be too complex to be helpful. While this is in keeping with APA format to provide descriptive stats of the raw data, this is not an APA journal. However, I leave this to editorial discretion.

7. PLOS authors have the option to publish the peer review history of their article (what does this mean?). If published, this will include your full peer review and any attached files.

Reviewer #1: No

Reviewer #2: Yes: Ming Wai Wan

---

## [Editor Report · Acceptance letter]

16 Apr 2020

PONE-D-19-29540R1 

I like you better when you are coherent.
Narrating autobiographical memories in a coherent manner has a positive impact on listeners’ social evaluations. 

Dear Dr. Vanaken:

I am pleased to inform you that your manuscript has been deemed suitable for publication in PLOS ONE. Congratulations! Your manuscript is now with our production department. 

With kind regards,

on behalf of

Dr. Sara Rubinelli 

Academic Editor

PLOS ONE